# Infrared Dim Small Target Detection Networks: A Review

**DOI:** 10.3390/s24123885

**Published:** 2024-06-15

**Authors:** Yongbo Cheng, Xuefeng Lai, Yucheng Xia, Jinmei Zhou

**Affiliations:** 1Key Laboratory of Science and Technology on Space Optoelectronic Precision Measurement, Chinese Academy of Sciences, Chengdu 610209, China; chengyongbo22@mails.ucas.ac.cn (Y.C.); xiayucheng@ioe.ac.cn (Y.X.); zhoujm@ioe.ac.cn (J.Z.); 2Institute of Optics and Electronics, Chinese Academy of Sciences, Chengdu 610209, China; 3University of Chinese Academy of Sciences, Beijing 100049, China

**Keywords:** infrared dim small target, deep learning, target detection network, public datasets, evaluation metrics

## Abstract

In recent years, with the rapid development of deep learning and its outstanding capabilities in target detection, innovative methods have been introduced for infrared dim small target detection. This review comprehensively summarizes public datasets, the latest networks, and evaluation metrics for infrared dim small target detection. This review mainly focuses on deep learning methods from the past three years and categorizes them based on the six key issues in this field: (1) enhancing the representation capability of small targets; (2) improving the accuracy of bounding box regression; (3) resolving the issue of target information loss in the deep network; (4) balancing missed detections and false alarms; (5) adapting for complex backgrounds; (6) lightweight design and deployment issues of the network. Additionally, this review summarizes twelve public datasets for infrared dim small targets and evaluation metrics used for detection and quantitatively compares the performance of the latest networks. Finally, this review provides insights into the future directions of this field. In conclusion, this review aims to assist researchers in gaining a comprehensive understanding of the latest developments in infrared dim small target detection networks.

## 1. Introduction

Infrared detection utilizes the difference between target radiation and background radiation for detection. Compared to visible light detection systems, infrared detection has the advantages of being less affected by weather, working day and night, and having strong anti-interference capabilities [1]. Therefore, infrared detection is widely used in fields such as maritime surveillance, maritime domain awareness, and guidance [2,3,4,5,6,7]. 

The detection of infrared small targets is often conducted at a relatively long distance from the target. In addition, there is atmospheric interference and the attenuation of infrared radiation caused by long-distance transmission media [8,9,10]. This results in challenges such as complex image backgrounds, small target imaging areas, and a low ratio of signal-to-noise in the detection process.

After decades of development, researchers have proposed many algorithms for detecting infrared dim small targets. Traditional detection algorithms are usually manually designed based on the characteristics of these targets and can be roughly categorized into three types: (1) Methods based on filtering [11,12,13,14], and background suppression models [10,15,16,17]. These methods have low computational complexity and low complexity, but they can only suppress uniform backgrounds to a certain extent and cannot solve the problem of low detection rates and poor robustness in complex backgrounds. (2) Methods based on the local contrast of the human visual system [4,18,19,20,21,22,23,24]. These methods are simple and easy to implement but suffer from the influence of significant non-target areas in the background and prominent edges, leading to poor detection performance. (3) Methods based on low-rank models [25,26,27,28,29,30,31]. These methods transform the target detection task into completing tasks with sparse low-rank tensors. However, they are typically time-consuming and have a higher false alarm rate in infrared images of dark targets. In summary, these traditional methods rely on manually crafted features, and require prior knowledge of the background scene. They are suitable for detecting targets in simple scenarios with stable and prominent features, but perform poorly in the face of complex and dynamic real-world scenarios.

The introduction of R-CNN in 2014 marked the first application of deep learning in the field of object detection [32]. Since then, deep learning-based object detection methods have been able to address a large number of object detection problems. In recent years, especially after researchers such as Wang et al. [33] and Dai et al. [34] released infrared small target datasets, more and more researchers have been incorporating deep learning algorithms into the field of infrared dim small target detection. They customize the design of deep learning networks based on the characteristics of infrared small target detection to improve detection performance. Currently, there are several articles summarizing traditional single-frame infrared dim small target detection methods [35,36,37,38,39]. However, the development of infrared dim small target detection methods based on deep learning is progressing rapidly, with new methods and networks constantly emerging; yet, literature reviews in this area are still relatively scarce. This review provides a comprehensive summary of the latest progress in deep learning networks used for infrared small target detection, categorizing them based on the six key issues in this field: (1) enhancing the representation capability of small targets; (2) improving the accuracy of bounding box regression; (3) resolving the issue of target information loss in the deep network; (4) balancing missed detections and false alarms; (5) adapting for complex backgrounds; and (6) lightweight design and deployment issues of the network. The contributions of this study are outlined as follows.

A survey and summary of the twelve publicly available infrared small target datasets were conducted.A classification summary of the latest deep learning methods for infrared dim small target detection was provided.An overview of existing loss functions and evaluation metrics for infrared small target detection was conducted.A comparison of the metrics for the latest infrared dim small target detection networks was performed.

## 2. Public Infrared Small Target Detection Datasets

A large amount of data forms the foundation of deep learning, and the lack of publicly available datasets for infrared dim small target detection has always been a barrier in related research. In recent years, some researchers have captured, synthesized, and annotated datasets while studying networks for infrared dim small target detection, making them publicly available for researchers to use. This review summarizes publicly available datasets of infrared dim small target detection. Details on these datasets are shown in Table 1. Examples of some real and synthetic images are shown in Table 2.

MFIRST

H. Wang et al. [33] released a large synthetic single-frame infrared small target detection dataset in addition to two datasets: one containing 11 real infrared sequences with 2098 frames, and another containing 100 individual real infrared images with different small objects. Additionally, the authors synthesized small targets using 2D Gaussian functions and overlaid them on infrared natural scene images to create a dataset. The MFIRST dataset comprises a total of 10,000 infrared images, with target sizes ranging from 6 × 6 to 20 × 20 pixels.

2.SIRST

Dai et al. [34] contributed a dataset called SIRST, which includes 427 annotated infrared images captured at short, medium, and 950 nm wavelengths. They provided five types of annotations for the images to support both detection and segmentation tasks. SIRST is the first publicly available real infrared dim small target dataset with high-quality images and labels. Later, Dai et al. released an improved version called SIRST-v2, which is specifically designed for single-frame infrared small target detection. It consists of 515 images selected from thousands of infrared sequences, representing different scenes.

3.IRSAT

Hui et al. [40] used a refrigerated mid-wave infrared camera to capture low-altitude flying small aircraft targets, contributing an infrared dim small target dataset. This dataset covers scenes including sky, ground backgrounds, and various scenarios, with a total of 22 segments of data, 30 flight tracks, 16,177 frames of images, and 16,944 targets. Each target corresponds to an annotated position, and each data segment corresponds to an annotation file.

4.IRSTD-1k

Zhang et al. [41] contributed a dataset captured with an infrared camera, which consists of one thousand 512 × 512 infrared images. They manually annotated the targets at the pixel level. The targets include drones, organisms, boats, and vehicles, while the backgrounds include oceans, rivers, fields, mountains, cities, and clouds. The dataset has significant clutter and noise.

5.SIRST-Aug

Zhang et al. [42] found in their research that differences in image sizes and data volumes limit the performance of networks, leading to overfitting and model convergence issues. Therefore, they performed cropping, rotation, and data augmentation on the SIRST dataset to create SIRST-Aug. The cropped image is 256 × 256 in size.

6.NUDT-SIRST

Li et al. [43] developed a large-scale synthetic infrared small target dataset called NUDT-SIRST, which consists of 1327 images. For the synthetic images, they first used a Gaussian kernel function and collected the target templates; then, they employed the adaptive target size function to ensure the size of the target and the combination of the target and the background was reasonable. Finally, they used the adaptive intensity function and Gaussian blurring function to adjust the target’s intensity and blur its boundary, respectively. In this dataset, approximately 37% of the images contain at least two targets; 27% of the targets occupy less than 0.01% of the entire image area; 96% of the targets meet the SPIE definition of small targets; and around 32% of the targets are located outside the top 10% of image brightness values. This dataset includes various target types, diverse target sizes, and different cluttered backgrounds, which present a more challenging scenario for infrared dim small target detection.

7.NCHU-SIRST

Shi et al. [44] contributed an NCHU-SIRST dataset including 590 infrared images, and most of these infrared frames are selected from 6300 real-world infrared images photographed by using a DL700 infrared camera. In total, 590 infrared images are divided into 273 training frames and 317 test frames. The target scene is roughly classified into six categories: architecture, cloudless sky, complex clouds, continuous clouds, sea, and trees. The targets contain aircraft, birds, and ships. The target sizes are distributed between 3 × 3 pixels and 9 × 9 pixels, which fully conforms with the SPIE definition.

8.Dataset fusion survey

Kou et al. [45] marked the anchor box of those as mentioned above five public infrared small target datasets (MFIRST, SIRST, SIRST-Aug, NUDT-SIRST, and IRSTD-1k), which include the number of targets, centroid coordinates, anchor box, and target pixel size.

9.IRST640

Chen et al. [46] synthesized an infrared small target dataset called IRST640, which consists of 1024 images with a size of 640 × 512. They generated one or more infrared small targets on real-world scene images, with background clutter including clouds, buildings, and trees.

10.SLR-IRST

Wang et al. [47] performed data collation and data extension on some existing infrared small target datasets. Then, they combined these with the Canny function and manual assistance to label the small target, obtaining a high-quality synthetic dataset called SLR-IRST. This dataset contains three kinds of labels.

11.IRDST

Sun et al. [48] built a dataset consisting of 40,650 frames of real infrared images and 102,077 frames of synthetic infrared images called IRDST, which contains three kinds of labels. The real images and the background of synthetic images are captured by a 7.5–13.5 μm FLIR camera in the DJI Zenmuse XT platform. The simulation dataset is formed of captured real backgrounds and targets simulated by a Gaussian simulation method.

## 3. Infrared Small Target Detection Network

Deep-learning-based object detection algorithms automatically extract features of objects, achieving significant success in the field of visible light target detection. In recent years, researchers have introduced deep learning methods into the field of infrared small target detection. However, due to the characteristics of infrared small targets, directly applying object detection networks often encounters some problems. Therefore, the networks need to be customized. After years of development, researchers have proposed some networks designed specifically for dim small target detection. Based on the following six key issues that the latest networks focus on, this chapter categorizes and summarizes the latest progress in the field of infrared dim small target detection networks: (1) enhancing the representation capability of small targets; (2) improving the accuracy of bounding box regression; (3) resolving the issue of target information loss in the deep network; (4) balancing missed detections and false alarms; (5) adapting for complex backgrounds; and (6) lightweight design and deployment issues of the network. In addition, infrared small target detection is divided into detection based on single-frame images and detection based on multi-frame images. This chapter focuses on summarizing the single-frame-based infrared small target detection networks.

### 3.1. Enhancing the Representation Capability of Small Targets

Feature extraction plays an important role in target detection processes. Features of infrared dim small targets are often sparse and constrained. To enhance the characterization capability of small targets, researchers have devised specialized feature extraction methods tailored for infrared dim small targets.

The method based on LCM primarily utilizes local information in the spatial domain [20], using the local contrast between image blocks and their neighborhoods as local features to construct saliency maps and segment small targets. Dai et al. proposed a novel model-driven deep network called ALC-Net [49], which combines networks and conventional model-driven methods. They integrated local contrast priors in convolutional networks and exploited a bottom–up attentional modulation to integrate low-level and high-level features. The architecture of the ALC-Net and its modules are shown in Figure 1.

Zhang et al. [50] proposed a multi-scale infrared small target detection method based on the combination of traditional methods and deep learning, achieving a good balance in background suppression and target extraction.

Yu et al. also combined networks and the local contrast idea to propose a novel multi-scale local contrast learning network (MLCL-Net) [51]. First, they obtained local contrast feature information and constructed the local contrast learning structure (LCL). Based on this, they built a multi-scale local contrast learning (MLCL) module to extract and fuse local contrast information at different scales. In [52], they proposed an attention-based local contrast learning network (ALCL-Net). They introduced the attention mechanism based on the LCL and proposed ResNet32 for feature extraction. In the feature fusion stage, they proposed a simplified bilinear interpolation attention module (SBAM). It not only speeds up the inference process and reduces pixel shifting, but also focuses on target features in the absence of context. The architecture of the ALCL-Net they proposed is shown in Figure 2.

Zhao et al. proposed an innovative gradient-guided learning network (GGL-Net) [53]. They also used LCL for local contrast learning. In the feature extraction section, they proposed the Gradient Supplementary Module (GSM) to encode the raw gradient information into deeper network layers and rationalize the embedding of attention mechanisms to enhance feature extraction. In addition, they proposed a two-way guided fusion module (TGFM) to promote multi-scale feature fusion. The structure of GGL-Net is shown in Figure 3.

The pyramid structure network has some loss of feature information during the dimensionality reduction operation of the convolutional layer, which results in a high rate of missed detection. To solve this problem, Bai et al. embedded a bottom–up pyramid in the feature pyramid network (FPN) [54], and proposed a cross-connected bidirectional pyramid network (CBP-Net), as shown in Figure 4. The double-pyramid structure preserves the shallow details of the small target.

Wu et al. proposed a deep interactive U-Net (DI-U-Net) [55]. It integrates a multi-level residual U-block that contains both long skip and short connections to keep the feature resolution.

Qi et al. proposed a fusion network architecture of transformer and CNN (FTC-Net) [56]. As shown in Figure 5, the CNN-based branch uses a U-Net with skip connections to obtain low-level local details. The transformer-based branch learns long-range contextual dependencies to enhance target features.

Hou et al. proposed a robust infrared small target detection network (RISTD-Net) [57]. They designed a feature extraction framework combining manual feature methods. Later, they proposed an Infrared Small-target Detection U-Net (ISTDU-Net) [58], which can convert the input image into a target probability likelihood map. ISTDU-Net introduces feature map groups in network downsampling and introduces a fully connected layer in hopping connections to improve small target feature characterization.

Wang et al. introduced a coarse-to-fine interior attention-aware network (IAA-Net), as shown in Figure 6 [59]. They designed a region proposal network (RPN) with ResNet18 as the backbone to propose coarse candidate target regions and then generated semantic feature maps. Finally, the attention encoder (AE) picked out the candidate regions on the semantic feature map.

Zhang et al. introduced an attention-guided pyramid context network (AGPC-Net) [42]. They designed a context pyramid module (CPM) that is better adapted to the characteristics of small infrared targets, resulting in performance gains. Among them, they proposed an attention-guided context block (AGCB) to estimate the correlation of pixels within and between patches and highlights targets. The architecture of the AGPC-Net is shown in Figure 7.

Ren et al. proposed a multi-scale Gaussian significance and attention feature fusion network (MGAF) [60]. They proposed the MGAF module and introduced the attention mechanism to extract the multi-scale Gaussian saliency features of small targets.

Zhou et al. proposed a deep low-order sparse patch image network [61], termed Deep-LSP-Net, which converts an infrared image into a patch image and then decomposes it into a superposition of low-order background components and sparse target components.

Wu et al. tailored a multi-branch topology for infrared small targets [62], using gradient information to extract edge features and a multi-branch structure to compensate for shape features.

Wang et al. proposed a pyramid-feature fusion target detection network [63] called RLPGB-Net, which combines reinforcement learning with targets to highlight the significant features of targets. Then, they introduced the boundary attention (GB) module, which can make full use of the information of the context and enhance the detection ability of infrared dark small targets.

Wang et al. proposed an effective Attention-Guided Feature Enhancement Network (AFE-Net) that introduces attention mechanisms in the encoding and decoding layers [64]. Non-local operations in different layers are cascaded to remove clutter similar to infrared target features.

### 3.2. Improving the Accuracy of Bounding Box Regression

Bounding box regression is a core task in object detection. Due to the characteristics of infrared dim small targets, conventional methods may not be suitable. Researchers have proposed some bounding box regression methods tailored for infrared dim small target detection, which are summarized in this section.

To achieve more accurate regression of infrared small target bounding boxes, Yang et al. introduced the Normalized Gaussian Wasserstein Distance (NWD) to measure the similarity between distributions with minimal overlap or no overlap [65], which is more suitable for infrared small targets than the IoU metric. Furthermore, they provided corresponding annotated versions of bounding boxes for the current public infrared small target datasets.

Li et al. applied CIoU to infrared dim small target detection for accurate bounding boxes regression, which takes into account the overlap area of bounding boxes [66], the distance between their centers, and the aspect ratio. Additionally, they used CIoU in Soft-NMS to obtain more accurate bounding box results.

Liu et al. utilize Distance Intersection Over Union (DIOU) for bounding box regression [67], taking into account scale, overlap, and the distance between targets and anchor boxes. This method directly minimizes the distance between two bounding boxes, improving the accuracy of regression.

Dai et al. proposed the one-stage cascade refinement (OSCAR) network for infrared small target detection that aims to perform cascaded bounding box regression [68]. In addition, they incorporated a NoCo branch to improve performance by suppressing low-quality predicted bounding boxes caused by pseudo boxes. The architecture of OSCAR is shown in Figure 8.

### 3.3. Resolving the Issue of Target Information Loss in the Deep Network

During the process of the networks becoming deeper, the features of dim and small targets are prone to being lost or overwhelmed by the background features. Additionally, loss of target information can also occur during the process of feature fusion. Researchers have proposed some methods to address this issue of information loss.

Tong et al. proposed an enhanced asymmetric attention (EAA) U-Net [69]. They presented an EAA module that uses both same-layer feature information exchange and cross-layer feature fusion to improve feature representation. EAAU-Net extracts and fuses the target feature maps in two stages, explicitly solving the problem of small targets being lost at deeper layers.

In 2015, He et al. proposed residual networks (ResNet) to solve the training problem of networks with too many layers [70]. ResNet can also be used to solve the problem of disappearing of small target features caused by deep layers. Ma et al. designed a feature extraction network consisting of two residual network modules [71], five convolutional modules, and an ASPP module to prevent feature loss. Some of the existing infrared small target detection methods usually use ResNet20 [51]. Yu et al. considered that the performance of ResNet20 is insufficient [52]. Then, they proposed ResNet32 by deepening ResNet20 in units of the stage, which learns the feature information of more scales while avoiding the loss of small target feature information caused by deepening the network layers of the stage.

Zhou et al. proposed U-Net++ [72], addressing the challenge of information loss when fusing low- and high-level feature maps in U-Net through densely nested convolution. Li et al. proposed a densely nested attention network (DNA-Net) [43], where they integrated multiple U-shaped sub-networks and established connections between the encoder and decoder sub-networks to enhance information retention, particularly for small targets within deep layers. In addition, they incorporated ResNet18 and attention modules to prevent information from being diluted. However, such a structure ensures detection accuracy but increases the complexity of the network. Bao et al. improved DNA-Net by retaining the densely nested attention network structure in Dense Nested Attention Network (DNA-Net) and introducing a Swin-transformer in the feature extraction stage to enhance feature continuity [73], resulting in better performance. Hu et al. designed an ISmall-Net with a multi-scale nested interaction module (MNIM) [74], which covers multiple U-shaped sub-networks to construct a densely nested structure. Compared to DNA-Net, MNIM features more node connections, facilitating enhanced preservation of information related to small targets. Figure 9 illustrates the U-shaped structures as discussed.

Inspired by DNA-Net, Chuang et al. proposed AMFU-net based on UNet3+ [75,76], which introduces an attention module to prevent gradient vanishing by applying residual blocks to the encoder and decoder.

Zhang et al. introduced CA-U2-Net, a refinement of the U2-Net tailored to make the network more focused on infrared dim and small targets [77]. By streamlining the top two coding and decoding layers to retain essential features, the modified model achieves a notable reduction in size while significantly enhancing accuracy.

Wu et al. proposed a simple yet efficient network, RepISD-Net [62], leveraging diverse network architectures with identical model parameters for both training and inference. This design ensures robust feature representation, mitigating the risk of target loss within deeper network layers.

Wu et al. introduced UIU-Net [78], a novel architecture that integrates a compact U-Net within a larger U-Net backbone. This innovative design prevents information loss for small targets during downsampling without the need for a classification backbone. By preserving object resolution and enhancing network depth simultaneously, this method offers a promising solution in target detection tasks.

### 3.4. Balancing Missed Detections and False Alarms

It is typically challenging to achieve both low missed detections and low false alarms for object detection networks; the same is true for infrared small target detection networks. It is important to research how to balance these two metrics.

Wang et al. proposed a deep adversarial learning framework, as shown in Figure 10 [33]. This framework disentangles the tasks of reducing missed detections (MD) and false alarms (FA) into two distinct subtasks. Through adversarial training of these two models, a balance between MD and FA is achieved.

Du et al. introduced the balancing precision and recall network (BPR-Net) aimed at balancing precision and recall through a unique multi-scale attention mechanism encompassing three key aspects [79]. The network utilizes an encoder–decoder framework for detecting infrared small targets, illustrated in Figure 11. Firstly, within the encoder, the scale fusion module integrates features from related images of varying resolutions. Secondly, in the decoder, the channel fusion module (CFM) amalgamates valuable information from multiple channels. Lastly, they incorporate the wavelet transform cross-layer skip layer (WTL) to bolster the interaction between the decoder layers.

In the MINP-Net proposed by Meng et al. [80], they introduced a noise prediction network to enhance the recognition of noise. Then, they planned a region localization branch (RPB) to predict the rough location of infrared small targets. This enables the proposed framework to achieve a good balance between missed detections and false alarms. The architecture of the proposed MINP-Net is shown in Figure 12.

### 3.5. Adapting for Complex Backgrounds

In some applications of infrared dim small target detection, the background is more complex, such as complex cities, cloudy skies, and forests. Such background images have a high variance in grey values and a low signal-to-noise ratio, which makes target detection more difficult. There are some studies on networks for complex backgrounds.

Yang et al. introduced a depth feature fusion infrared network (DFFIR-net) and two methods to solve the infrared small target detection problem in the complex background [81], and the framework of the proposed network is shown in Figure 13. They used a smoothing operator to acquire smooth features of small targets, significantly enhancing the small targets in the obtained smoothness image while effectively suppressing the background. To address the issue of the sensitivity of the smoothing operator to isolated noise, they designed the integrated detection framework DFFIR-net. By leveraging the strong learning capability of deep learning, this framework more fully explores the original features and smooth features of small targets. It utilizes a multi-layer feature fusion mapping network to fuse shallow features from two branch networks of feature extraction networks in a layered manner, enriching the feature representation of small targets and suppressing background clutter. Ma et al. designed a multi-layer joint upsampling strategy to map small targets and suppress background [71]. During the upsampling process, the feature mapping network can effectively suppress background clutter through convolution operations in the joint upsampling model. In the GLFM-net model, they used six joint upsampling models to construct a multi-layer joint upsampling network. Through this network, they obtained a small target feature map image where background clutter was completely suppressed, achieving infrared small target detection in complex backgrounds. In addition, Ma and Yang et al. proposed a multi-scale 2D Gaussian label generation strategy that can improve detection performance under small training samples.

Zhang et al. introduced novel infrared shape network (IS-Net) [41]. First, they designed an edge block inspired by Taylor finite difference to enhance the edge information to improve the contrast between target and background, which improves the detection performance of the network in complex backgrounds. Then, they applied a bottleneck structure to remove high-frequency noise in infrared images.

In complex backgrounds, the infrared signals of small targets are weak, making them more susceptible to being overwhelmed. This leads to higher rates of missed detections and false alarms. Shi et al. proposed an infrared small target detection method using coordinate attention and feature fusion to cope with the abovementioned problems, named CAFF-Net [71]. Firstly, they proposed a deep and shallow feature fusion strategy. The feature fusion network they introduced concatenates and merges low-level structural and texture features with high-level semantic features to reduce the missed detection rate of small infrared targets. Then, they connected the coordinate attention module to the main network to enhance target saliency and suppress background interference in the feature maps, thus significantly reducing the false alarm rate of infrared small target detection in complex scenes.

Zhang et al. proposed a CA-U2-Net by improving U2-Net to address the problem of infrared small target detection and shape retention in complex backgrounds [77]. Specifically, they designed top–down attention blocks, which utilize the feature maps of the encoding layer and the corresponding decoding layer of the next layer modulated by top–down attention as the input to the decoding layer. This can suppress parts in the model learning process that are irrelevant to the red targets, thereby reducing the missed detection rate in complex scenes.

### 3.6. Lightweight Design and Deployment Issues of the Network

Infrared dim small target detection networks usually need to be deployed on edge devices; so, the real-time performance of the network is very important, which requires simple network structures and low computational overhead. However, improving network detection performance often requires a more refined algorithm design, resulting in complex network structures and increased computational efforts. Balancing the performance and real-time capability is one of the most important issues in infrared dim small target detection. Some studies have been proposed to improve the real-time capability of networks while meeting detection performance requirements.

Hu et al. introduced ST-Net to improve detection performance in complex background [82]. Meanwhile, they chose a binarized model with the highest memory gain and acceptable performance degradation in small target detection tasks for hardware acceleration. First, they employed a computational transformation strategy for better hardware implementation. Then, they designed a dedicated hardware architecture for this network. A single infrared image was divided into 16 × 16 pixels blocks as the basic processing object for reusability. Finally, they design a parallel processing architecture to improve the parallelism of computation and meet the requirements of real-time applications in realistic scenarios. Their proposed accelerator achieves a detection speed of 56 FPS under CMOS 28 nm technology, with power consumption as low as 48.7 mW.

Chuang et al. used a full-size jump connection based on UNet3+ as a basis to avoid a dense nested structure [75,76], introducing AMFU-Net. It reduces the computational cost by fusing features with a small number of parameters and achieves lightweight. The parameters of AMFU-Net are 2.17 MB, and the AMFU-Net achieves a detection speed of 29.5 FPS.

Ma et al. proposed an extremely lightweight infrared dim small target detection network, MiniIR-net, as shown in Figure 14 [83]. The model size of the MiniIR-net is only 40 KB. First, they proposed a multi-scale target context feature extraction (TCVE) module to reduce the number of parameters required for model fitting. Then, they designed a feature mapping upsampling network by fusing the deep and shallow features to improve the feature mapping capability. The size of their proposed network is 0.039 Mb.

Wu et al. proposed a simple yet efficient network (RepISD-Net) [62]. The network uses identical model parameters for both training and inference, which balance the efficiency and performance of the network simply.

Kou et al. introduced lightweight an IR small target segmentation network (LW-IRST-Net) [84]. To improve the computational efficiency, they discarded the feature fusion module and developed a new lightweight encoding and decoding structure, which achieves lightweight while achieving good segmentation performance. The parameters and FLOPs of LW-IRST-Net are only 0.16 M and 303 M, respectively. In addition, they designed a post-processing module that enhances the robustness of the application deployment and can meet the requirements of real-time, high-precision, online dynamic target feature adjustment. 

## 4. Loss Function and Evaluation Metrics for Infrared Dim Small Target Detection

Loss functions are used to optimize the parameters of a model and directly impact the performance of the model on training data, while evaluation metrics are used to measure the performance of the model on test data or in practical applications. Evaluation metrics are typically the performance indicators that users care about, while the loss function is the objective function that optimization algorithms focus on. Both evaluation metrics and loss functions are important components in object detection tasks. This chapter summarizes the commonly used loss functions and evaluation metrics for the detection of infrared small targets.

### 4.1. Loss Function

The choice of loss function is crucial for the detection network. In the infrared small target detection task, positive and negative samples are unbalanced. Researchers have proposed various loss functions to solve this problem, and this section summarizes these loss functions commonly used for infrared small target detection.

#### 4.1.1. BCE Loss

The binary cross-entropy is used to evaluate the goodness of binary classification model predictions, which measures the difference between the probabilities output by the sigmoid function and the true labels. Its definition is as follows:$$L_{BCE}=-\frac{1}{N}\sum_{i=1}^{N}y_{i}*\log\left(p\left(y_{i}\right)\right)+\left(1-y_{i}\right)*\log\left(1-p\left(y_{i}\right)\right)$$
where $p\left(y_{i}\right)$ is the probability of the label $y_{i}$, and N is the number of samples. Sometimes, a weight value is added to improve training effectiveness.

#### 4.1.2. Dice Loss

The Dice coefficient is a metric used to evaluate the similarity between two samples, where a higher value indicates greater similarity between the two samples. The Dice loss and the Dice coefficient sum up to one. The Dice loss has significant applications in semantic segmentation problems. Its definition is as follows:$$L_{Dice}=1-\frac{2\left|X\cap Y\right|}{\left|X\right|+\left|Y\right|}$$
where X represents the pixel labels of the actual segmented image, while Y represents the pixel categories of the segmented image predicted by the model. The Dice loss can alleviate the negative impact of foreground–background imbalance in samples. Training with Dice loss focuses more on target regions, but it suffers from loss saturation issues. Therefore, using Dice loss alone often does not yield satisfactory results. It needs to be combined with other losses, such as Binary Cross-Entropy (BCE) loss, for better performance.

#### 4.1.3. Soft-IoU Loss

Soft-IoU loss is developed based on IoU but with smoother handling to better optimize the training process. Compared to traditional IoU loss, Soft-IoU loss can better handle class imbalance situations. Its definition is as follows:$$L_{SoftIoU}=\frac{\sum_{i,j}P_{i,j}*Y_{i,j}}{\sum_{i,j}\left(P_{i,j}+Y_{i,j}-P_{i,j}*Y_{i,j}\right)}$$
where $Y\left(i,j\right)$ denotes the true mask label and $P\left(i,j\right)$ denotes the predicted score map obtained by the network.

#### 4.1.4. MSE Loss

The pixel-by-pixel mean square error loss function (MSE) represents the average of the squared differences between predicted values and true values. Its definition is as follows:$$L_{MSE}=\frac{\sum_{i=1}^{I}\sum_{j=1}^{J}{\left(P\left(i,j\right)-Y\left(i,j\right)\right)}^{2}}{I*J}$$
where $Y\left(i,j\right)$ is the training label and $P\left(i,j\right)$ is the prediction image of the network output, and I and J are the scale parameters of the training image.

### 4.2. Evaluation Metrics

Different evaluation metrics correspond to different aspects of network performance. The following are surveyed on the existing evaluation metrics used for infrared dim small target detection.

#### 4.2.1. Precision and Recall

In infrared small target detection, precision and recall are the most fundamental and commonly used evaluation metrics, defined by the confusion matrix. The confusion matrix for a binary classification problem consists of four numbers, as shown in Figure 15. The definitions of precision and recall are as follows:$$precision=\frac{TP}{TP+FP}$$
$$recall=\frac{TP}{TP+FN}$$

In general, precision and recall are interrelated; when precision is high, recall is low, and when recall is high, precision is low. The F1score is the harmonic mean of precision and recall, providing a better reflection of the model’s performance, and it is defined as follows:$$F1score=\frac{2*precision*recall}{precision+recall}$$

F-measure is the weighted harmonic mean of recall and precision under nonnegative weight, which adds weight values for precision. It is defined as follows:$$F_{measure}=\frac{\left(\beta^{2}+1\right)precision*recall}{\beta^{2}precision+recall}$$
where $\beta^{2}$ = 0.3, generally.

#### 4.2.2. Pd and Fa

In infrared small target detection, probability of detection (Pd) and false alarm rate (Fa) are commonly used metrics to evaluate the detection performance of networks. Pd represents the proportion of correctly detected targets among all detected targets, and it is defined as follows:$$Pd=\frac{N_{correct}}{N_{all}}$$
where $N_{correct}$ and $N_{all}$ are the number of correctly detected targets and all targets, respectively.

In reference [79], the authors define Pd at the pixel level. It is defined as follows:$$Pd=\frac{TP}{TP+FP}$$
where TP represents the correct pixel count of the detected targets. FN and FP denote the numbers of pixels that misclassify targets as backgrounds and pixels that misidentify backgrounds as targets, respectively.

False alarm rate (Fa) measures the ratio of falsely predicted pixels over all image pixels, and it is defined as follows:$$Fa=\frac{P_{false}}{P_{all}}$$
where $P_{false}$ and $P_{all}$ represent the numbers of falsely predicted pixels and all image pixels, respectively.

#### 4.2.3. IoU

Intersection over Union (IoU) calculates the intersection and union ratio between the predicted border and the actual border, and it is defined as follows:$$IoU=\frac{A_{inter}}{A_{all}}$$
where $A_{inter}$ and $A_{all}$ represent the interaction areas and the union areas, respectively. In some research on infrared small target detection, especially when emphasizing semantic segmentation, $A_{inter}$ and $A_{all}$ are also represented as the interaction pixels and the union pixels, respectively. In addition, the Mean Intersection over Union (mIoU) is the average IoU of the model for each type of prediction result.

nIoU is the normalization of IoU, and it is defined as follows:$$nIoU=\frac{1}{N}\sum_{i}^{N}\frac{TP\left[i\right]}{T\left[i\right]+P\left[i\right]-TP\left[i\right]}$$
where N is the total number of samples, $TP\left[\cdot \right]$ represents the number of true positive pixels, and $T\left[\cdot \right]$ and $P\left[\cdot \right]$ represent the number of ground truth and predicted positive pixels, respectively.

#### 4.2.4. SCR

In the field of infrared small target detection, signal-to-clutter ratio gain (SCRG) and background suppression factor (BSF) are important evaluation indexes. SCRG and BSF could be used to verify the target enhancement ability and background suppression ability of different methods. 

SCR measures the signal-to-clutter ratio of an image, and it is defined as follows:$$SCR=\frac{\left|\mu_{t}-\mu_{b}\right|}{\sigma_{b}}$$
where $\mu_{t}$ is the target pixel mean, $\mu_{b}$ is the background region pixel mean, and $\sigma_{b}$ is the standard deviation of the background pixel value. 

In addition, when an image contains multiple targets, the average SCR is commonly used to evaluate the difficulty of multi-target detection and the performance of the method. It is defined as follows:$$\bar{SCR}=\frac{1}{N}\sum_{i}^{N}SCR\left[i\right]$$
where N represents the number of targets and i represents the order of targets.

SCRG reflects the enhancement degree of target input and output relative to the background, and can also be used to describe the difficulty of small target detection. It is defined as follows:$$SCRG=\frac{{SCR}_{out}}{{SCR}_{in}}$$
where ${SCR}_{in}$ and ${SCR}_{out}$ are the signal-to-clutter ratios of the input and output images, respectively.

BSF reflects the effect of background suppression, and it is defined as follows:$$BSF=\frac{C_{in}}{C_{out}}$$
where $C_{in}$ is the standard deviation of the input image and $C_{out}$ is the standard deviation of the output image.

#### 4.2.5. PR and ROC

In the field of infrared small target detection, the Receiver Operating Characteristic Curve (ROC) utilizes Fa as the horizontal axis and Pd as the vertical axis. The area of the graph formed by the ROC curve and the X-axis can be used as a comprehensive measure, namely AUC (Area Under Curve); the larger this area, the better the performance of the method. In addition, Wang et al. pointed out that it is meaningless whether a detection result is achieved with a high detection rate but a high false alarm rate [59]; so, they used the middle part of the ROC curve to calculate the AUC (from false alarm rate 10^−4^ to 10^−2^) and normalize the AUC values into [0, 1]. 

Precision Recall Curve (PR) utilizes recall as the horizontal axis and precision as the vertical axis, which can comprehensively reflect the recall and precision of the model. The closer the curve to the upper-right corner, the better performance of the algorithm. Average precision (AP) represents the average value of the model under various recall scenarios, corresponding to the area under the PR curve.

The ROC curve and PR curve are both standards used to measure the classification performance of models, but the PR curve is more sensitive to the sample proportion.

#### 4.2.6. FLOPs and FPS

Floating point operations (FLOPs) represent the number of floating point operations in a network and can measure the complexity of a network. Network parameters refer to the total number of parameters that need to be trained in a network model and can measure the size of the model. For a detection network, FLOPs and network parameters can reflect the network’s requirements for hardware computational power and memory while ensuring detection performance.

Frames Per Second (FPS) is a metric that gauges the speed of a network. It assesses the detection speed by measuring the number of images processed per second and is directly correlated with hardware performance.

## 5. Quantitative Comparison of Network Performance

This chapter compares the detection performance and computational complexity of the latest surveyed infrared dim small target detection networks. The experimental data for 20 recent infrared dim small target detection networks previously mentioned are summarized in Table 3, with all networks evaluated on either a portion or the entirety of the SIRST public dataset. Researchers conducted experiments on the ACM method [34] and ALC-Net, which are earlier outstanding infrared dim small target detection networks, using various proportions of training, validation, and test sets. When MFIRST was used as the training set, the IoU metric notably underperformed compared to others. Regarding the IoU and mIoU metrics, networks with attention mechanisms such as IS-Net and GGL-Net, as well as contrast-based ALCL-Net, MLCL-Net, and LSP-Net, demonstrated superior performance. For the Pd metric, networks like FTC-Net, IS-Net, DNA-Net, RepISD-Net, and UIU-Net exhibited outstanding performance, with RepISD-Net and UIU-Net achieving a Pd value of up to 100% on the SIRST dataset.

The parameters and FLOPs of networks are presented in Table 4. Networks like IAA-Net, DNA-Net, AGPC-Net, and UIU-Net exhibit more node connections or nested structures, prioritizing detection accuracy but leading to higher computational complexity, which often fails to meet real-time requirements. On the other hand, LW-Net, CAFF-Net, and RepISD-Net emphasize lightweight design, making them more hardware-friendly. In LW-Net and IS-Net, the authors further designed accelerators to achieve better real-time performance.

## 6. Summary and Future Outlooks

Firstly, this review summarizes currently available public datasets for infrared dim small target detection. Then, this review focuses on infrared dim small target detection networks from the past three years, categorizing them based on the key issues they addressed. Researchers have delved into these issues from various aspects, including target feature representation, bounding box regression, feature maintenance, and background suppression, balancing missed detections and false alarms, as well as lightweight design, achieving a significant amount of research results. Finally, this review summarizes the existing loss functions and evaluation metrics for assessing the performance of infrared dim small target detection networks and provides a quantitative comparison of the latest networks’ performance.

The accuracy of infrared dim small target detection has been significantly improved based on deep learning methods. However, there are still some problems that obstruct the development and application of infrared dim small target detection. Therefore, we discuss and suggest future development directions in this section:Although some researchers have captured or synthesized certain infrared small target datasets and made them publicly available, there is still a demand for large-scale and diverse datasets that are suitable for engineering applications. For instance, the scarcity of datasets with extremely low target radiation intensity has led to a shortage of algorithms for dim small target detection; the categories of backgrounds need to be subdivided to improve the detection accuracy in specific application scenarios. In addition, compared with single-frame images, streaming video sequences can provide more motion information and can be applied for target tracking tasks. Therefore, establishing large-scale, diverse, and video sequence-based datasets remains an essential foundational task.Regarding dataset annotation issues, how to accurately annotate ground truth at the pixel level is an important issue that researchers need to pay more attention to in the future. On the one hand, researchers should consider the effects of atmospheric, optical system’s point spread function and phase noise of pixel discrete sampling when annotating images. On the other hand, researchers might investigate innovative methods to automatically identify mislabeled annotations and mitigate these effects during the training process.Deep learning is suitable for detecting targets with unstable and inconspicuous features in known scenarios. However, their interpretability is dimer compared to that of traditional methods. Currently, some studies combine these two methods [34,50,51], inserting modules based on traditional methods into the network to complete certain sub-tasks. In the future, enhancing the deeper fusion of the two methods may lead to breakthroughs in this field.Single-band infrared dim and small target images lack color, texture, and distance information. Fusion tasks between infrared and other detection approaches (multi-spectral bands, multiple detectors, radar, etc.) can provide extra information such as spectral information, high-resolution texture, and target distance, thereby obtaining more comprehensive target information. So, enhancing the application of information fusion in infrared dim target detection may provide new insights in this field.Currently, most of the existing research tends to focus on the detection performance of the networks. These networks often come with complex structures. However, infrared dim small target detection networks usually need to be deployed on resource-constrained edge devices. Therefore, real-time performance and detection performance are equally important. In the future, how to balance the two performances is a considerable issue for researchers to focus on.

## Figures and Tables

**Figure 1 sensors-24-03885-f001:**
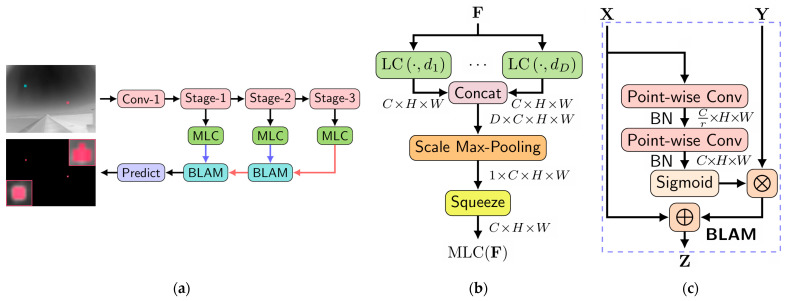
(**a**) Architecture of the ALC-Net. The network backbone is modified based on ResNet-20, consisting of three stages. (**b**) The same-layer multi-scale local contrast (MLC) module. In the figure, F represents the input feature map, $LC\left(\cdot ,d_{i}\right)$ denotes the module measuring multi-scale local contrast, where its input · and $d_{i}$ correspond to the input feature map and various dilation rates, respectively; (**c**) The cross-layer bottom–up local attentional modulation (BLAM) module. In the figure, X represents the low-level features, Y represents the high-level features, and z denotes the fused multi-scale local contrast feature map ([49] Figures 3 and 4) Copyright © 2021, IEEE.

**Figure 2 sensors-24-03885-f002:**
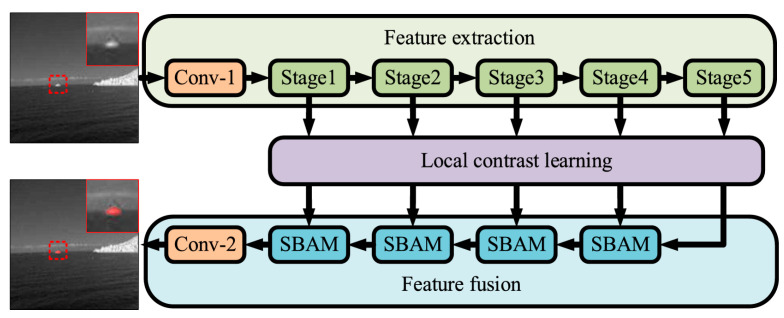
Architecture of the ALCL-Net using the proposed ResNet32, which consists of Conv-1 and stages 1–5 ([52] Figure 1) Copyright © 2022, IEEE.

**Figure 3 sensors-24-03885-f003:**
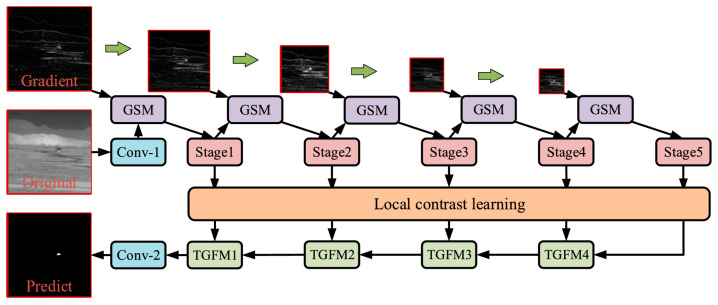
The structure of the GGL-Net. Conv-1, conv-2, and stages 1–5 in this figure are consistent with those in ALCL-Net mentioned above ([53] Figure 2) Copyright © 2023, IEEE.

**Figure 4 sensors-24-03885-f004:**
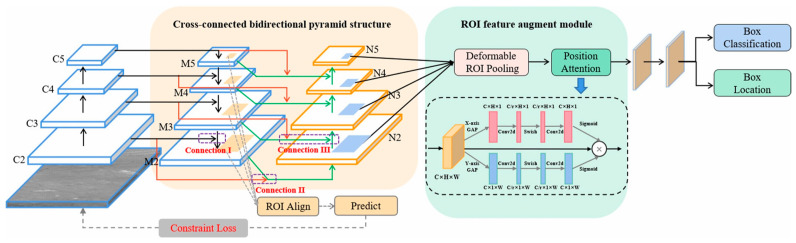
The framework of the CBP-Net consists of three strategies: the main cross-connected bidirectional pyramid structure, the ROI feature augment module, and the Regular constraint loss ([54] Figure 1) Copyright © 2022, IEEE.

**Figure 5 sensors-24-03885-f005:**
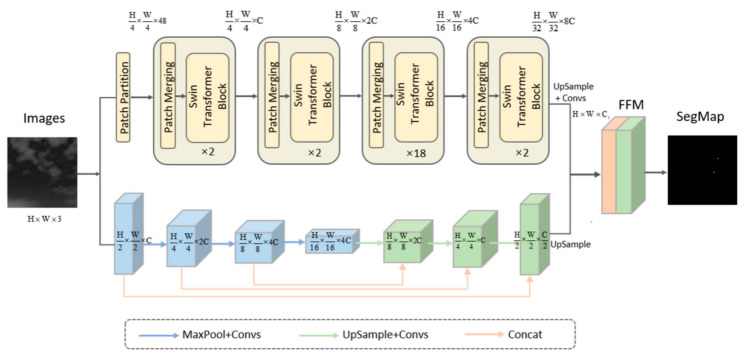
Architecture of the FTC-Net. The top half is the CNN-based branch. The bottom half is the transformer-based branch ([56] Figure 2). This content is licensed under the “CC BY-SA 4.0” license. To view this license, visit https://creativecommons.org/licenses/by-sa/4.0/ (accessed on 15 January 2024).

**Figure 6 sensors-24-03885-f006:**
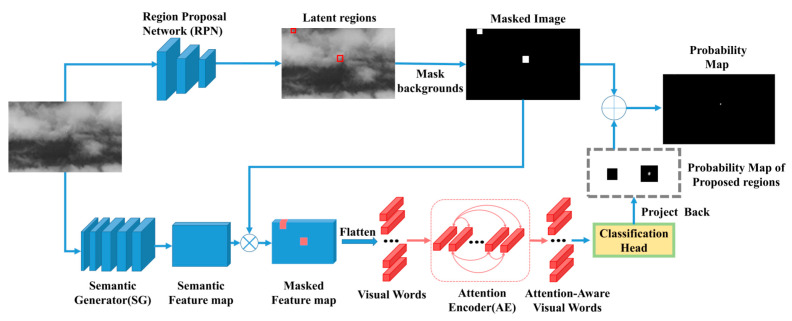
Architecture of the IAA-Net, which is composed of an RPN, an SG, and an AE. For the input image, RPN proposes coarse local regions, SG generates semantic feature maps. Then, candidate regions on the semantic feature map are picked out and flattened into visual words and encoded by AE. Finally, probabilities are predicted by visual words ([59] Figure 2) Copyright © 2022, IEEE.

**Figure 7 sensors-24-03885-f007:**
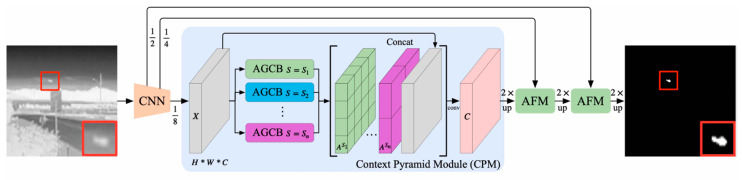
Architecture of the AGPC-Net. The AFM is their proposed asymmetric fusion module ([42] Figure 1) Copyright © 2023, IEEE.

**Figure 8 sensors-24-03885-f008:**
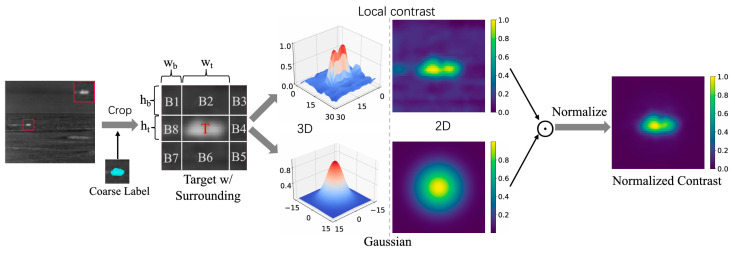
Generation of NoCo map. First, the linear local contrast is calculated to provide a distribution that matches the appearance of the target. Then, a Gaussian is applied to give preference to the geometric center of the target. Finally, a coarse label is used to semantically normalize the labeled target region and some background pixels, and the rest of the values are set to 0. This process results in an NoCo map that is not sensitive to disturbances in the bounding box, making it a reliable representation of the target ([68] Figure 4) Copyright © 2023, IEEE.

**Figure 9 sensors-24-03885-f009:**
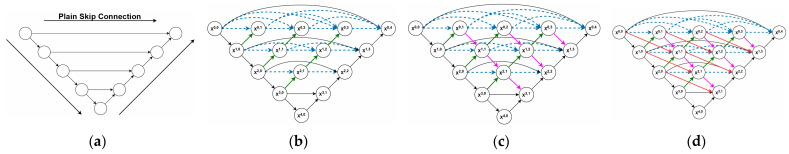
The illustration of the U-shape structures. These arrows indicate connections and point the way. (**a**) U-Net; (**b**) U-Net++; (**c**) DNA-Net; (**d**) ISmall-Net. ([74] Figure 2) Copyright © 2023, IEEE.

**Figure 10 sensors-24-03885-f010:**
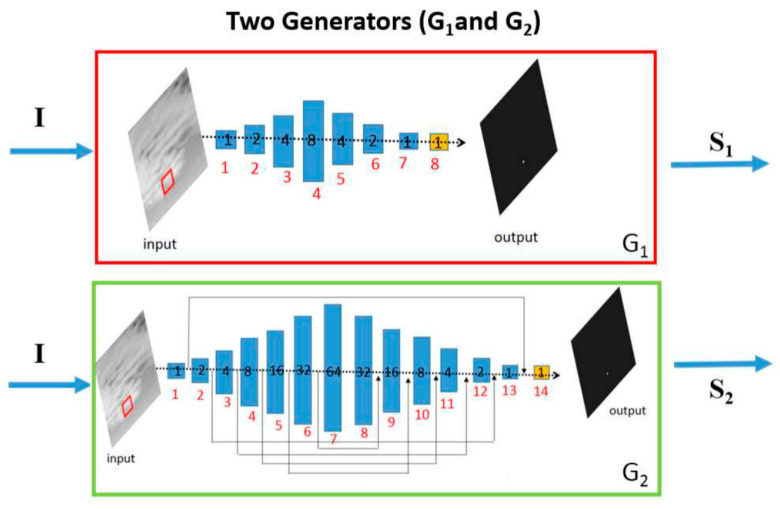
The overview of the deep adversarial learning framework ([33] Figure 2) Copyright © 2019, IEEE.

**Figure 11 sensors-24-03885-f011:**
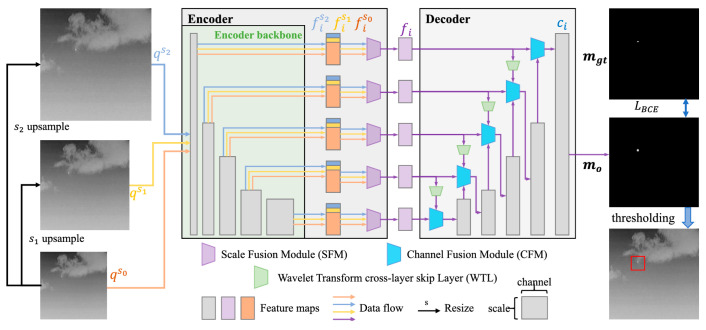
The structure of BPR-Net. The network employs an encoder–decoder framework for detecting infrared small targets. The encoder comprises a shared encoding backbone and a set of parallel and unshared SFMs. The decoder employs a bottom–up architecture, including a series of decoder blocks, each of which contains a CFM and a WTL ([79] Figure 2) Copyright © 2023, IEEE.

**Figure 12 sensors-24-03885-f012:**
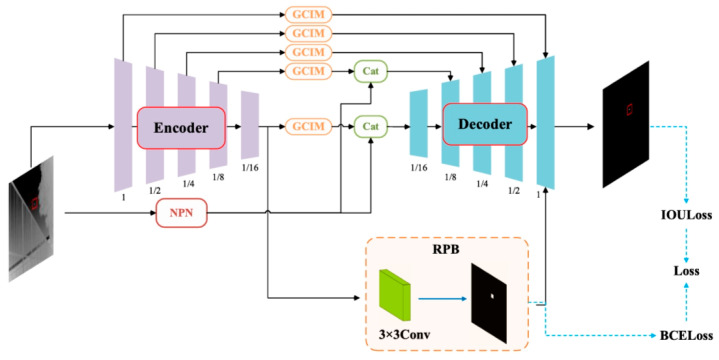
Architecture of the proposed MINP-Net, where GCIM indicates the proposed GCIM, NPN indicates the noise prediction network, RPB represents the regional positioning branch, and Cat denotes the concatenated operation ([80] Figure 1) Copyright © 2023, IEEE.

**Figure 13 sensors-24-03885-f013:**
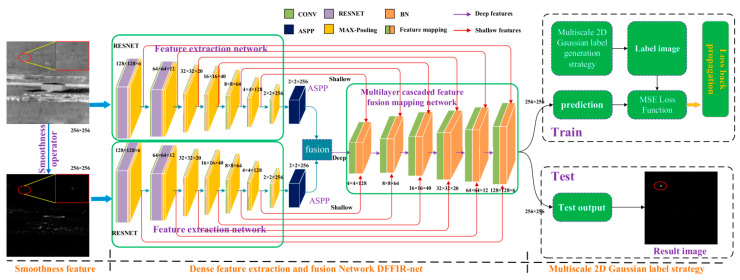
Detection framework of the DFFIR-Net ([81] Figure 2) Copyright © 2022, IEEE.

**Figure 14 sensors-24-03885-f014:**
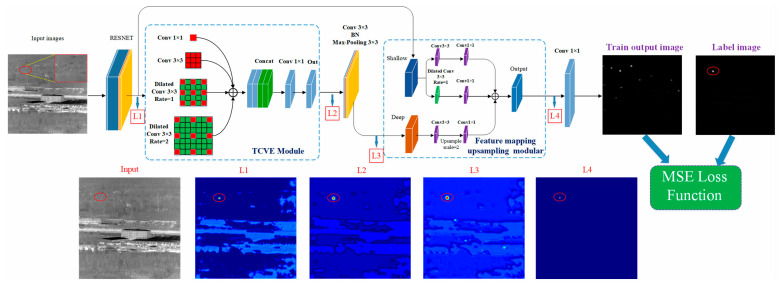
The illustration of the MiniIR-net model. L1, L2, L3, and L4 are visual images of the characteristic matrix of the corresponding layer in the MiniIR-net ([83] Figure 1) Copyright © 2023, IEEE.

**Figure 15 sensors-24-03885-f015:**
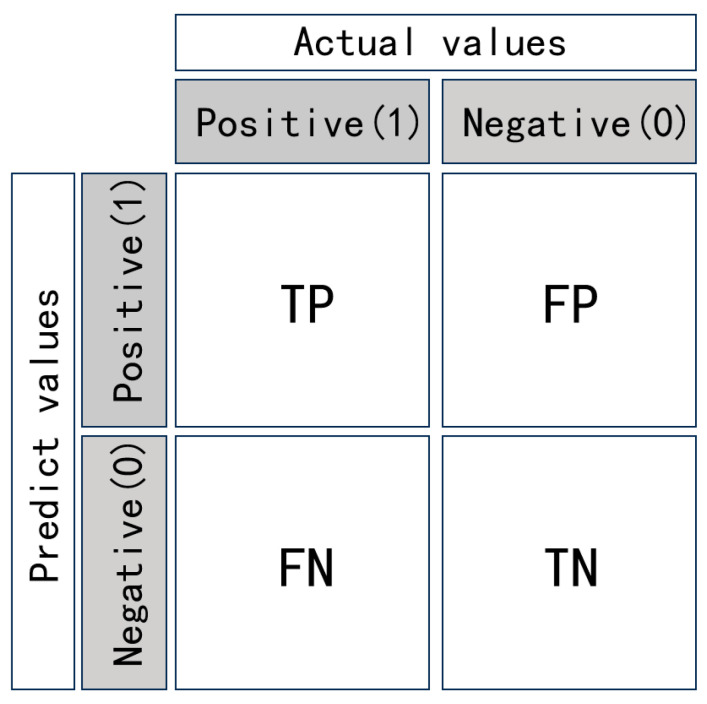
Illustration of the binary confusion matrix.

**Table 1 sensors-24-03885-t001:** Details on the present public infrared small target datasets.

Datasets	Image Type	Image Num	Provided Label	Background	Target Size	Image Size
MFIRST	Synthetic	10,000	Pixel	Cloud/City/Sea	6 × 6–20 × 20	173 × 98–640 × 480
IRSAT	Real	16,177	Center	Sky/Ground	3 × 3–9 × 9	256 × 256
SIRST	Real	427	Pixel/box	Cloud/City/River/Road	2 × 2–14 × 34	96 × 135–400 × 592
SIRST V2	Real	515	Pixel/Box	Cloud/sky/City/Mountain/Field/Road	5 × 5–20 × 20	268 × 202–1024 × 1024
SIRST-Aug	Real	9070	Pixel/Box	Cloud/City/River/Road	5 × 5–20 × 20	256 × 256
IRSTD-1K	Real	1000	Pixel	Cloud/City/Sea/River/Mountain/Field	1 × 1–56 × 33	512 × 512
NUDT-SIRST	Synthetic	1327	Pixel	Cloud/City/Sea/Field	3 × 3–9 × 9	256 × 256
NCHU-SIRST	Real	590	Pixel	Cloud/City/Tree/Sea	3 × 3–9 × 9	256 × 256
Dataset fusion survey ^1^	Real/Synthetic	21,898	Box/Center	Cloud/City/River/Road	1 × 1–56 × 33	96 × 135–640 × 480
IRST640	Synthetic	1024	Pixel	Cloud/Building	1 × 1–9 × 9	640 × 512
SLR-IRST	Real/Synthetic	2689	Pixel/Box/Center	Cloud/Building/River/Lake/ Tree	1 × 1–14 × 34	256 × 256
IRDST	Real/Synthetic	142,727	Pixel/Box/Center	Cloud/Tree/Lake/Building	1 × 1–9 × 9	720 × 480/ 934 × 696

^1^ In the dataset fusion survey, the authors provided only the labels; the rest of the data are a concatenation of five datasets.

**Table 2 sensors-24-03885-t002:** Examples of some real and synthetic images.

Datasets		Examples of Datasets			
Real	SIRST	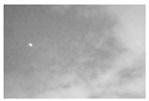	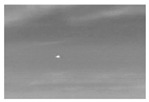	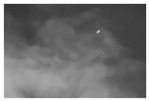	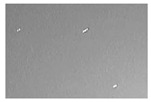
	IRSTD-1K	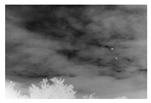	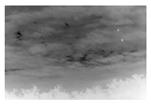	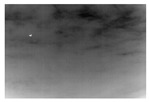	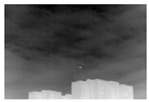
	NCHU-SIRST	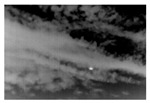		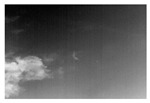	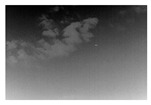
Synthetic	MFIRST	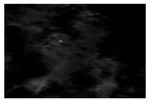	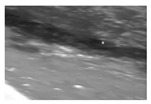	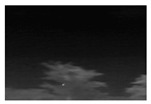	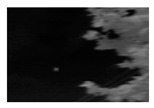
	NUDT-SIRST	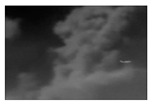	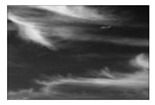	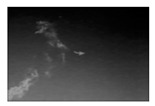	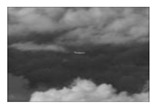
	IRST640	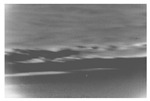	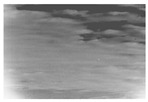	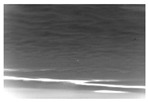	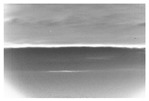

**Table 3 sensors-24-03885-t003:** Comparison results of the latest networks. The test refers to the proportion of the SIRST dataset used for testing.

Network	Test	IoU	nIoU	Pd	Fa (10^−6^)	Data Reference
MDvsFA [33]	50%	0.6030	-	0.8935	56.35	[43]
	5%	0.4114	0.5653	-	-	[78]
	20%	0.603	-	0.8935	56.35	[62]
ACM [34]	30%	0.743	0.731	0.9391	-	[34,64]
	20%	0.7331	0.7227	0.9391	-	[56]
	20%	0.7233	0.7143	0.9633	9.325	[41]
	50%	0.7033	-	0.9391	3.728	[43]
	5%	0.6178	0.6378	-	-	[78]
ALC [49]	30%	0.757	0.728	0.9657	-	[49,64]
	20%	0.7570	0.7280	0.9657	-	[56]
	20%	0.7431	0.7312	0.9734	20.21	[41]
	50%	0.7333	-	0.9657	30.47	[43]
EAAU [69]	30%	0.771	0.746	-	-	[69]
DI-U [55]	23%	0.762	0.743	-	-	[55]
FTC [56]	20%	0.7772	0.7702	0.9905	-	[56]
IS [41]	20%	0.8002	0.7812	0.9918	4.924	[41]
ISTDU [58]	20%	0.5883	-	0.8991	40.63	[62]
IAA [59]	20%	0.655	0.6722	0.9087	41.43	[80]
MLCL [51]	30%	0.772	0.755	-	-	[51]
ALCL [52]	30%	0.792	0.774	-	-	[52]
AGPC [42]	30%	0.6490	0.6481	-	-	[61]
DNA [43]	50%	0.775	-	0.9848	2.353	[43]
	30%	0.6746	0.6399		-	[61]
	20%	0.7747	-	0.9848	2.35	[62]
GGL [53]	22%	0.814	0.786	-	-	[53]
RepISD [62]	20%	0.7781	-	1.0	4.22	[62]
UIU [78]	5%	0.7825	0.7515	-	-	[78]
	20%	0.7825	-	1.0	6.39	[62]
Deep-LSP [61]	30%	0.8243	0.8185	-	-	[61]
MINP [80]	20%	0.7508	0.7318	0.9620	3.07	[80]
AFE [64]	30%	0.774	0.752	0.999	-	[64]

**Table 4 sensors-24-03885-t004:** Computational complexity comparison.

Network	Size	Parameters (M)	FLOPs (G)	Data Reference
ACM [34]	256 × 256	0.52	1.72	[62]
MDvsFA [33]	128 × 128	3.77	370.67	[80]
EAAU [69]	480 × 480	2.07	-	[69]
ISTDU [58]	256 × 256	2.76	29.76	[62]
IS [41]	256 × 256	1.09	121.90	[62]
IAA [59]	256 × 256	14.05	875.69	[80]
DNA [43]	256 × 256	4.70	56.08	[62]
AGPC [42]	-	12.35	43.18	[84]
AMFU [76]	256 × 256	2.17	-	[76]
LW [84]	-	0.1632	0.303	[84]
UIU [78]	256 × 256	50.54	217.84	[62]
RepISD [62]	256 × 256	0.28	25.76	[62]
MINP [80]	256 × 256	15.73	26.30	[80]
AFE [64]	-	2.07	1.67	[64]

## Data Availability

The data presented in this study are available in this article.

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
