# Peer review of "Infrared Dim Small Target Detection Networks: A Review"

_sensors, 2024, doi:10.3390/s24123885_

Round 1

Reviewer 1 Report

Comments and Suggestions for Authors

This paper comprehensively summarizes recent progress in infrared small target detection, including public datasets, detection networks, and evaluation metrics. The rich and detailed references are beneficial for researchers to gain a comprehensive understanding of the latest developments in infrared weak small target detection networks. Regarding the specific content of the paper, the following suggestions are proposed:

1. Infrared small target detection is divided into detection based on single-frame images and detection based on multi-frame images. The review does not seem to emphasize which type of image detection it is targeting. Although there are fewer networks for multi-frame detection, this aspect should be clarified.

2. The classification of five methods in the paper, particularly the delineation between the first and fourth categories, appears somewhat ambiguous. It is recommended to provide clearer explanations of the classification in section 3.4 and elaborate on how the networks mentioned in comparison to others further adapt to complex backgrounds.

3. The choice of loss function is crucial for detection networks. Various loss functions have been proposed for balancing positive and negative samples in IR small target detection networks, which could be supplemented in the review.

4. Could the conclusion section consider the current issues with datasets? With an increasing number of algorithm researchers contributing to public datasets in this field, it is worth discussing whether algorithms based on existing datasets are applicable in engineering. For example, the target radiation intensity in dataset images is generally significantly higher than the background, indicating a scarcity of datasets specifically focusing on dim small targets.

5. Regarding dataset annotation issues, the imaging results of targets are inevitably influenced by the atmospheric and optical system's point spread function. How to accurately annotate ground truth at the pixel level and the feasibility of further improving metrics based on numerous pixel-level networks in this context could also be discussed.

Author Response

Thank you very much for taking the time to review this manuscript. For the responses, please see the attachment.

Reviewer 2 Report

Comments and Suggestions for Authors

This review paper comprehensively summarizes public datasets, the latest deep neural networks, and evaluation metrics for infrared dim small target detection. This paper is interesting but I would suggest modifications before its final publication:

1.     In the last paragraph in Introduction, authors mentioned “… categorizing them based on the five key issues in this field”. Authors should present the five key issues clearly in this paragraph.

2.     For better understanding for readers, examples of several public dataset images need to be shown and compared. By doing this, the similarity between synthetic and real images can be observed.

3.     In various datasets, the methods to generate synthetic dim targets and background components are different. Authors should add the explanation on various methods to generate synthetic images.

4.     Authors need to add the structural explanation of block diagrams, and explain figure symbols in detail. For example, In Fig. 1, why stage-1 to stage-3 are considered? What does LC( \dot, d1) mean? What are X , Y, Z ? Authors should also include detailed descriptions of other figures in the manuscript.

5.     For the task of dim target detection, the classification mechanism and the generation of accurate bounding boxes make significant impacts on performance results. In Section 3, authors should include these key issues.

Author Response

Thank you very much for taking the time to review this manuscript. For the detailed response, please see the attachment.

Round 2

Reviewer 1 Report

Comments and Suggestions for Authors

The authors have successfully addressed most of the issues, thus I suggest to accept this manuscript for appearing.

I think you may misunderstand my fifth suggestion. I would like you to consider and discuss the fifth suggestion in the conclusion, as most of the datasets you presented in Table 1 are pixel-level labels. You have done an excellent job in Section 3.2, but as this is a review paper, I recommend retaining the original, more reasonable classification method (five key issues).

Author Response

Response to Reviewer  Comments:

       Thank you very much for providing suggestions on our manuscript  again. We  agree with your viewpoints. Accordingly, we have made the following modifications:

1.We have made some minor modifications to the first discussion in the conclusion section.

  • Although some researchers have captured or synthesized certain infrared small target datasets and made them publicly available, there is still a demand for large-scale and diverse datasets that are suitable for engineering applications. For instance, the scarcity of datasets with extremely low target radiation intensity has led to a shortage of algorithms for dim small target detection; the categories of backgrounds need to be subdivided to improve the detection accuracy in specific application scenarios. In addition, compared with single-frame images, streaming video sequences can provide more motion information and can be applied for target tracking tasks. Therefore, establishing large-scale, diverse, and video sequence-based datasets remains an essential foundational task.

2.We have added a section in the conclusion discussing the issues related to dataset annotation.

  • Regarding dataset annotation issues, how to accurately annotate ground truth at the pixel level is an important issue that researchers need to pay more attention to in the future. On the one hand, researchers should consider the effects of atmospheric, optical system’s point spread function and phase noise of pixel discrete sampling when annotating images. On the other hand, researchers might investigate innovative methods to automatically identify mislabeled annotations and mitigate these effects during the training process.

In addition, we greatly appreciate your affirmation of our work. However, another reviewer raised concerns regarding the issue of bounding box regression, which is a critical aspect of object detection tasks. Therefore, we would like to retain the newly added key issue (3.2).

Reviewer 2 Report

Comments and Suggestions for Authors

The manuscript has been well revised for each comment. 
